# Investigation of the Effect of Palmaris Longus Presence on the Upper Extremity and Hand Functions in Individuals of Different Ethnic Origins

**DOI:** 10.3390/diagnostics15141763

**Published:** 2025-07-12

**Authors:** Onur Seçgin Nişanci, Rıdvan Yildiz, Sidrenur Aslan Kolukisa

**Affiliations:** 1Vocational School of Health Services, Department of Physiotherapy, Artvin Coruh University, Artvin 08000, Turkey; 2Atatürk Vocational School of Health Services, Department of Medical Services and Techniques, Dicle University, Diyarbakır 21280, Turkey; ridvanyildiz2023@gmail.com; 3Vocational School of Health Services, Department of Occupational Therapy, Artvin Coruh University, Artvin 08000, Turkey; sidrenuraslan@artvin.edu.tr

**Keywords:** palmaris longus, Georgian population, Turkish population, hand function, upper extremity

## Abstract

**Objectives**: This study aims to determine the presence rate of the palmaris longus (PLM) muscle in Turkish and Georgian individuals and to examine its effect on hand/upper extremity function. **Methods**: The study, conducted at Artvin Çoruh University, included 400 volunteer students (800 hands). The presence of PLM was evaluated with Schaeffer and Thompson tests. Upper extremity and hand functions were measured by the Upper Extremity Function Index (UEFI-15) and the Sollerman Hand Function Test. Data were evaluated with the chi-square test, Mann–Whitney U, and Linear Mixed Model analysis in the SPSS v22 program. **Results**: PLM muscle was significantly more common in Georgian individuals compared to Turkish individuals (*p* < 0.05). This difference was statistically significant, particularly in Georgian men (*p* < 0.05). There was no significant correlation between the presence of PLM and hand and upper extremity functions (*p* > 0.05). **Conclusions**: Although the PLM muscle shows different and significant prevalence rates in relation to ethnicity, it does not show a significant effect on hand function. Ethnicity is a determining factor in anatomical and surgical studies of the PLM, and its minimal functional impact on hand function makes its graft use safe.

## 1. Introduction

The palmaris longus (PLM), the superficial flexor muscle of the forearm in the upper extremity, is one of the most common variations and phylogenetically retrogressive muscles in humans [1]. This muscle has undergone numerous morphological changes over evolutionary time and can be observed in agenetic, paired, split, incomplete, and digastric forms [1,2]. These variations in the palmaris longus muscle are important in clinical radiological imaging and interventional surgery. This variety of variations can affect material adequacy in tendon graft planning, and can also cause pressure on the median nerve due to its proximity to the nerve. However, in some cases, it can also be misinterpreted as a tumor or mass [3,4]. Originating from the medial epicondyle, this tendinous muscle is located medial to the flexor carpi radialis muscle and lateral to the flexor carpi ulnaris muscle, and is frequently preferred as a graft in tendon transfer procedures [5,6]. The palmaris longus muscle, located between two flexor muscles, is often used in tendon transfers due to its sufficient tendon length, proximity to the surface, and low functional capacity [7]. Due to the numerous flexor muscles and tendons in the forearm region, the PLM plays a supportive role in grip strength and hand functions [8]. Therefore, its increasing use in reconstructive surgeries highlights the importance of understanding its anatomical variations across populations [9,10].

The PLM, which follows a pattern of phylogenetic regression, shows variation in prevalence among different ethnic groups. In large-scale studies conducted on the Turkish population, the absence of the palmaris longus in at least one hand was detected in 64% of the participants [11,12]. In a study conducted on the Ghanaian population, the absence of the PLM muscle was observed at a rate of 3.1%. This is below the average absence rate of 15% reported in the literature and suggests a lower or slower phylogenetic regression tendency among individuals of Ghanaian origin [13]. In a study conducted in the United States, which has a highly multi-ethnic population, the absence of the PLM muscle was found to be 13–14% among White and Latin American individuals, while it was reported at 3–4% among individuals of Asian and African descent [14]. Various studies have been conducted in the United States to investigate the prevalence of the palmaris longus muscle. In one study on this subject, the absence rate of the muscle was reported to be 4.5% in African Americans and 14.9% in White people. This situation demonstrates the need to take ethnic origin into account in surgical operations [15]. A review of other studies in the field reveals that most evaluations have focused on the presence or absence of the PLM and its effect on grip strength; however, assessments addressing overall hand and upper extremity functions have been found to be limited [6,10,11]. Studies on the palmaris longus muscle have generally focused on the presence or absence of the muscle, and its effect on grip strength, contribution to fine motor skills, and functionality in daily life have not been sufficiently investigated [7,8]. The study was designed based on the hypothesis that the palmaris longus muscle would be found in different ethnic groups at different prevalences. In this context, our study aims to determine the prevalence of the palmaris longus muscle in university students of Turkish and Georgian ethnic origin and to evaluate the effects of this variation on hand and upper extremity functions.

## 2. Materials and Methods

This cross-sectional descriptive study was conducted among students of Artvin Çoruh University and was approved by the Scientific Research and Publication Ethics Committee of Artvin Çoruh University, Republic of Türkiye (Approval No: E-18457941-050.99-140904; date: 4 July 2024). Informed consent forms were obtained from all participants prior to the study, and each individual completed a personal information form.

In the classification of participants according to ethnic groups, individuals who reported Georgian ancestry in the personal information form, specifically through information such as the birthplace of grandparents or family origin, and who could fluently speak or understand Georgian, were classified within the Georgian ethnic group. Participants who declared that they understood only Georgian were played a short Georgian audio recording and then asked simple comprehension questions to objectively assess their language proficiency. Conversely, individuals who only spoke Turkish fluently were classified in the Turkish ethnic group. Although this method is widely preferred in field research where direct ethnicity identification tools such as genetic analysis cannot be used, it remains a limited approach to defining ethnicity. In our study, the classification of participants by ethnic background—particularly those identified as Georgian—was based on information such as the birthplace of grandparents and proficiency in the Georgian language. However, we acknowledge that this approach may introduce misclassification bias, especially in culturally blended or multi-ethnic populations. Proficiency in Georgian does not necessarily confirm biological or cultural affiliation. To minimize this risk, we collected multidimensional data, including not only language proficiency but also ancestral background, sense of cultural identity, and self-identification. The classification process was conducted by integrating these criteria holistically. For future studies, we recommend the use of standardized ethnic identity questionnaires and, where possible, sensitivity analyses to strengthen the validity of ethnic classifications [16].

### 2.1. Inclusion Criteria

-Being 18 years of age or older;-Being enrolled at Artvin Çoruh University;-Having no history of congenital anomalies or previous trauma that could alter the anatomy of the hand and upper extremity.

### 2.2. Exclusion Criteria

-Having a history of congenital anomalies, surgical interventions, fractures, dislocations, or deformities related to the upper extremity;-Presence of neurological, rheumatological, or musculoskeletal disorders that could affect the functional use of the hand.

### 2.3. Assessment of the Palmaris Longus Muscle

The presence of the palmaris longus (PLM) muscle in participants was evaluated through clinical examination conducted by physicians and physiotherapists with a minimum of five years of professional experience in the field. The Schaeffer’s test was used to assess the PLM muscle. During the test, participants were asked to bring the wrist into slight flexion while opposing the thumb and the fifth finger. The PLM tendon was observed over the flexor retinaculum during this maneuver. If the tendon was not visibly apparent and could not be detected through palpation, the absence of the PLM was recorded. Each hand was tested twice, and in cases of uncertainty, a third repetition was performed. The Thompson test was also used to support the evaluation when needed [17] (Figure 1).

### 2.4. Functional Assessment

Upper Extremity—Short Form (UEFI-15): To assess the functional status of the upper extremity, the Upper Extremity Functional Index—Short Form (UEFI-15) was used. This form consists of 15 items based on activities of daily living and was completed by each participant only once, reflecting the combined use of both upper limbs. The assessment relies on the participant’s subjective evaluation of their upper extremity function. Each participant completed the UEFI-15 once, reflecting the functional status of both upper extremities as a whole. Based on the bilateral evaluation of the presence or absence of the PLM muscle, individuals were classified into unilateral or bilateral absence groups [18].

Sollerman Hand Function Test: The Sollerman Hand Function Test is a quantitative assessment used to evaluate hand functionality in activities of daily living. It consists of 20 different tasks, each scored between 0 and 4, with a total score ranging from 0 to 80. During the test, identical environmental conditions were maintained for all participants, and distracting stimuli were eliminated. The test was administered and scored separately for each hand. To prevent fatigue or decreased attention, a 5 min break was provided between assessments of the two hands. This approach ensured the reliability of the measurements [19]. Although a Turkish validity and reliability study of the Sollerman Hand Function Test has not been formally published, the original version of the test was applied in its standardized form by trained assessors. Instructions were conveyed verbally in Turkish while preserving the original task structure and scoring criteria. A specially designed functional panel was used to simulate various tasks included in the test (Figure 2). This panel incorporated components such as locks, zippers, laces, keys, door handles, and fastening mechanisms, enabling the standardized and reproducible execution of activities of daily living.

### 2.5. Sample Size and Power Analysis

An a priori power analysis was conducted using G*Power software (version 3.1) to determine the minimum sample size required for a chi-square test comparing two independent groups (Turkish and Georgian participants). Assuming a small effect size (Cohen’s w = 0.15), an alpha level of 0.05, and a statistical power of 0.80, the required sample size was calculated to be approximately 346 participants.

### 2.6. Statistical Analyses

Statistical analyses were performed using SPSS 22.0. A *p*-value < 0.05 was considered significant. The chi-square test (χ^2^) was used to compare the presence or absence of the palmaris longus (PLM) muscle across categorical variables such as sex and ethnic origin. For multiple comparisons between groups, post hoc chi-square analyses were conducted. To assess the normality of the data, the Kolmogorov–Smirnov test was applied, as the sample size exceeded 50 participants. For variables that did not follow a normal distribution, the Mann–Whitney U test was used to compare two independent groups.

Since the data related to the right and left upper extremities were obtained from the same individuals, hand-based data were treated as statistically dependent measurements. Therefore, to analyze the effect of the palmaris longus (PLM) muscle on hand function, a Linear Mixed Model (LMM) was employed. In the LMM, the Sollerman Hand Function Test score was set as the dependent variable; PLM presence, sex, and ethnic origin were defined as fixed effects; and participant identity (ID) was included as a random effect.

## 3. Results

A total of 400 volunteer university students (206 males, 194 females) participated in the study. The mean age of the participants was 21.4 ± 2.0 years. Based on ethnic origin, 212 individuals were of Turkish origin and 188 were of Georgian origin. Each participant’s right and left hands were evaluated separately, resulting in a total of 800 hands assessed. Each participant’s right and left hands were evaluated separately, resulting in a total of 800 hands included in the analysis. Although it is acknowledged that the hands of the same individual are not entirely statistically independent, they were treated as independent units in the analyses. This should be considered a limitation when interpreting the results (Table 1).

When the presence of the palmaris longus muscle was compared by ethnic origin, it was found to be significantly more common among individuals of Georgian origin than those of Turkish origin (*p* < 0.05) (Table 2).

According to the results of the post hoc chi-square analysis conducted to determine the source of the significant difference found in the chi-square test, the presence of the palmaris longus muscle was significantly higher in Georgian males compared to other sex and ethnic origin subgroups (*p* < 0.05) (Table 3).

When UEFI-15 scores were compared according to whether palmaris longus muscle absence was unilateral or bilateral in individuals of Turkish and Georgian origin, no significant difference was found in both groups (*p* > 0.05) (Table 4).

According to the results of the Linear Mixed Model analysis, neither the presence of the palmaris longus (PLM) muscle nor being of Georgian ethnic origin had a statistically significant effect on the Sollerman Hand Function Test scores (*p* > 0.05), although both showed a positive trend. Similarly, sex and hand side variables were not found to significantly affect hand function (*p* > 0.05). The intercept value of the model was calculated as 77.24, which represents the estimated mean test score for a Turkish female participant without a PLM muscle of the right hand (Table 5).

## 4. Discussion

The primary finding observed in this study is that the palmaris longus muscle (PLM) is more prevalent in individuals of Georgian origin compared to those of Turkish origin, with a statistically significant difference in terms of ethnic background. This difference was particularly more pronounced among Georgian male participants in subgroup analyses based on sex. However, it is important to distinguish between statistical significance and clinical or surgical relevance. Although this ethnic variation may be noteworthy from an anatomical or population-based perspective, its practical implications remain unclear. Specifically, the findings do not demonstrate whether this difference impacts surgical decision-making, tendon graft selection, or clinical outcomes. Thus, the observed ethnic disparity in PLM prevalence, while statistically valid, should be interpreted with caution in terms of clinical applicability. Furthermore, the presence of the PLM muscle did not show a significant effect on upper extremity function. Although both the presence of the PLM muscle and Georgian ethnicity appeared to have a positive effect on hand function, these associations were not statistically significant. These findings suggest that there is no direct relationship between the presence of the PLM muscle and functional performance; however, ethnicity may be a determining factor in the anatomical prevalence of the muscle. Additionally, the lack of a significant functional contribution to the hand supports the PLM muscle’s role as a safe and suitable candidate for grafting in nerve and tendon reconstruction procedures. An unexpected result of the study was that the dominant hand did not show a significant advantage in Sollerman Hand Function Test scores. This may be attributed to the homogeneous nature of the sample, consisting solely of young and healthy individuals, and the possibility that the test tasks were not sensitive enough to capture subtle differences related to hand dominance.

The PLM muscle is one of the most variative muscles in the human body in terms of anatomical diversity, and variables such as ethnicity and sex are important beyond individuality in the evaluation of its absence. Studies have shown that PLM variation is not only random but may be closely related to genetic and geographical characteristics [20,21,22]. In a multi-ethnic study of medical students in Malaysia, the presence of the PLM muscle was evaluated according to sex, ethnicity, and hand side. The Schaffer test showed that the highest rate of absence was observed in Indian students (28.2%), followed by Chinese (10.1%) and Malay (10.9%) students. It is interesting to note that the Indian group had the highest absence rate despite representing the smallest ethnic group in the study. No significant results were found in relation to hand side and sex, indicating the association of PLM with ethnicity [23]. Similarly, in another study conducted on Palestinian medical students, the absence rate of PLM was found to be 32%. Within this rate, bilateral absence was found to be 15.7% and unilateral absence was found to be 16.3%. One of the most interesting aspects of the study is that the absence of PLM was significantly more common in women than in men. Accordingly, absence in one or both hands was significantly higher in women than in men, with 40%, absence in both hands with 20.7%, and absence in the left hand only with 12%. This suggests that the presence of PLM may be related not only to ethnicity but also to sex [24]. In a study of 562 individuals in the Guilan region of northern Iran, the overall absence rate of PLM was reported to be 13.2% and no significant results were found according to hand side and sex. Although not significant, the higher rate of absence on the left side compared to the right side suggests that the hand side should not be ignored in regional variations [25]. In another study conducted in India, the relationship between the absence of the PLM tendon and weakness of the flexor digitorum superficialis muscle in the fifth finger was investigated, and unilateral absence of the PLM tendon was found to be 16.9% and bilateral absence was found to be 3.3% in Indian individuals. In this study, the absence of the PLM muscle was not significantly associated with sex and side of the hand, but was significantly associated with weakness of the flexor digitorum superficialis. According to sex, this was especially significant in males but not in females. This suggests that although the absence of the PLM does not cause a functional impairment in the hand alone, it may have clinical significance when considered with other anatomical variations. This should be taken into consideration, especially in surgical planning and functional anatomical evaluations [22]. In this study of a multi-ethnic sample of 516 individuals in the United States, the absence rate of the palmaris longus (PLM) muscle was assessed according to ethnicity, sex, and body side. The absence of the PLM muscle was not significantly associated with sex and body side, but was significantly associated with ethnicity. Accordingly, the rate of PLM absence was 2.9% in Asian individuals, 4.5% in African Americans, 14.9% in Caucasian individuals, and 13.1% in Hispanic white individuals, and these differences were statistically significant. The findings reveal once again that PLM muscle absence rates are influenced not only by individual, but also by ethnic-based genetic and embryologic, differences. In addition, it emphasizes that ethnicity should be taken into account when assessing the availability of the PLM muscle in tendon graft planning [14]. In a multi-ethnic study of 450 individuals in Malaysia, a Southeast Asian country, unilateral absence of the PLM muscle was found in 6.4% and bilateral absence in 2.9%. Among ethnic groups, absence of the PLM tendon was found in 11.3% of Malays, 10.7% of Indians, and 6.0% of Chinese, with significant differences observed. These data once again confirm that PLM absence may show ethnic-based differences and reveal that ethnic diversity may be determinant on this variation even within the Asian continent. The fact that the absence of the PLM muscle is more common in Malay and Indian individuals should be considered as another issue to be taken into consideration in surgical planning [26]. Furthermore, in a localized ethnic study of 600 individuals from the Yoruba population in Nigeria, the overall absence rate was 6.7%, with unilateral and bilateral absence rates of 5.4% and 1.5% in males and 6.0% and 0.4% in females, respectively. These data suggest that PLM absence rates may be lower in African populations, particularly in the Yoruba ethnic group, compared to European and Asian populations, supporting evidence that PLM variations can produce marked ethnic-based differences. Another remarkable finding of the study was the observation that in one case, the PLM tendon differentiated from the flexor carpi radialis muscle. This morphological observation supports the hypothesis that PLM and FCR muscles may originate embryologically from the same muscle mass. Therefore, this finding provides important embryological data not only on variation but also on muscle development [17]. In a study conducted to investigate the familial inheritance and genetic transmission of PLM in addition to variables such as ethnicity, sex, and hand side, a total of 82 individuals from 20 families were examined. The absence of PLM was observed in 59.75% (49 individuals) and the presence of PLM was detected in 40.25% (33 individuals). In the analysis of familial inheritance, 16 families (80%) were considered to have familial and possibly dominant inheritance with the observation of muscle presence or absence in at least one child consistent with the parents in the family. Furthermore, the absence of PLM in 20% (four families) of skipped generations suggests the role of the gene, penetrance, and variable expression. Investigations revealed that the muscle is not compatible with exogenous or recessive inheritance patterns, but is most likely associated with an autosomal dominant inheritance pattern. This suggests that the absence of the PLM muscle may not be genetically related to a single factor, but may be explained by various modulator genes and expression variability or, in some cases, mutational effects [27]. Collectively, these studies suggest that the presence of PLM muscle is a result of ethnic, sex, and possibly hereditary influences rather than a random difference. The wide range of PLM variations reported may be attributed to both non-standardized clinical examination techniques and sampling differences. This suggests that PLM should be evaluated on a population rather than an individual basis before being used as a graft in surgical applications.

In recent years, not only have anatomical variations been evaluated in studies on the PLM muscle, but also clinical studies have been conducted to determine the potential effect of the variative condition on the upper extremity and hand functions, especially grip strength and dexterity, and its functional value. In a study conducted at Bolu Abant İzzet Baysal University, 101 medical students were evaluated, and the absence rate of the PLM tendon was found to be 16.8% in the right arm and 17.8% in the left arm. There was no statistically significant difference between the presence of PLM and grip strength and wrist proprioception [28]. In a study conducted on athletes to examine the presence of PLM and its effect on hand function, although there was no statistical significance between the presence of PLM and grip strength in male athletes, a positive and significant relationship was found between the presence of PLM in the left hand and grip strength in female athletes. This finding suggests that the presence of PLM in female athletes may have a limited side-specific effect on hand function, but in general, the presence of PLM is not a determinant of functional performance [16]. In another study of 533 healthy volunteers, the effects of variations in the PLM and the flexor digitorum superficialis muscle (FDSM) in the fifth finger on the grip and pinch strength of the hand were evaluated. There was no significant difference between the presence of PLM tendon and grip strength, but it was observed to have a limited effect on pinch strength. This suggests that the PLM tendon is not critically important for general hand function, but may play a limited role in delicate grasping functions such as pinching [29]. In a study conducted on 240 medical students studying at Sultan Qaboos University in Oman, the relationship between the presence of PLM and grip strength was examined. The mean value was 30.84 ± 11.71 kg in individuals with PLM muscle and 35.05 ± 12.44 kg in those without PLM muscle; however, this difference was not statistically significant. This suggests that the PLM muscle does not have a direct determinant effect on hand function, especially grip strength, and that its functional load on the hand is minimal, and surgical grafting of this structure will not lead to a significant loss in terms of individual grip strength or overall hand function [6]. In the light of all these studies, it is concluded that the PLM muscle does not have a significant functional effect on the hand, but it is supportive especially in fine motor skills such as pinching, so its use as a nerve and tendon graft in surgeries does not result in a permanent loss of hand functions, and therefore it can be used safely. These results may be due to the fact that the PLM muscle is not the primary muscle for fine motor movements of the hand such as grasping and pinching, it has a superficial and thin tendon due to its structure, and superficial and deep strong flexor muscles can easily compensate for the functions of this muscle in the absence of PLM.

The fact that the sample group in the study population consists of only a certain age range and healthy individuals and is selected from a limited geographical region limits the generalizability of the results obtained. In addition, the fact that the hand tests were performed only over a cross-sectional period of time ignored the effects of possible variables in the hand function of individuals on the scores or prevented a long-term follow-up. In addition, the limitations of the study include the lack of imaging methods in addition to the determination of the presence of PLM with only clinical examination tests, and the lack of consideration of environmental factors that may affect functional capacity, such as cultural or occupational hand use habits of individuals.

## 5. Conclusions

This study revealed that the presence of the PLM muscle may vary according to ethnic origin and that Georgian individuals have a higher incidence than Turkish individuals. No significant correlation was found between the presence of the PLM muscle and hand function. The findings indicate that the PLM muscle can be safely used in nerve and tendon graft surgeries. Further studies with larger, multicenter clinical sample groups and the use of imaging methods are needed to understand the effects of variables such as ethnic origin, sex, age, and occupational hand use on the prevalence of the PLM muscle and hand function.

## Figures and Tables

**Figure 1 diagnostics-15-01763-f001:**
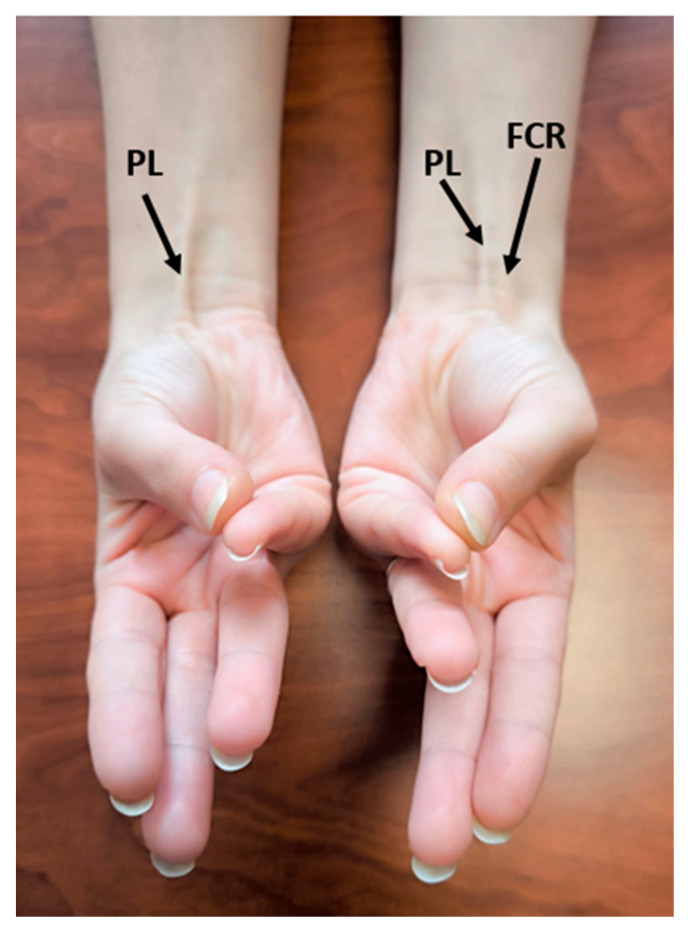
Schaeffer’s test demonstrating bilateral palmaris longus presence in a 21-year-old Georgian female participant (PL: palmaris longus, FCR: flexor carpal radial).

**Figure 2 diagnostics-15-01763-f002:**
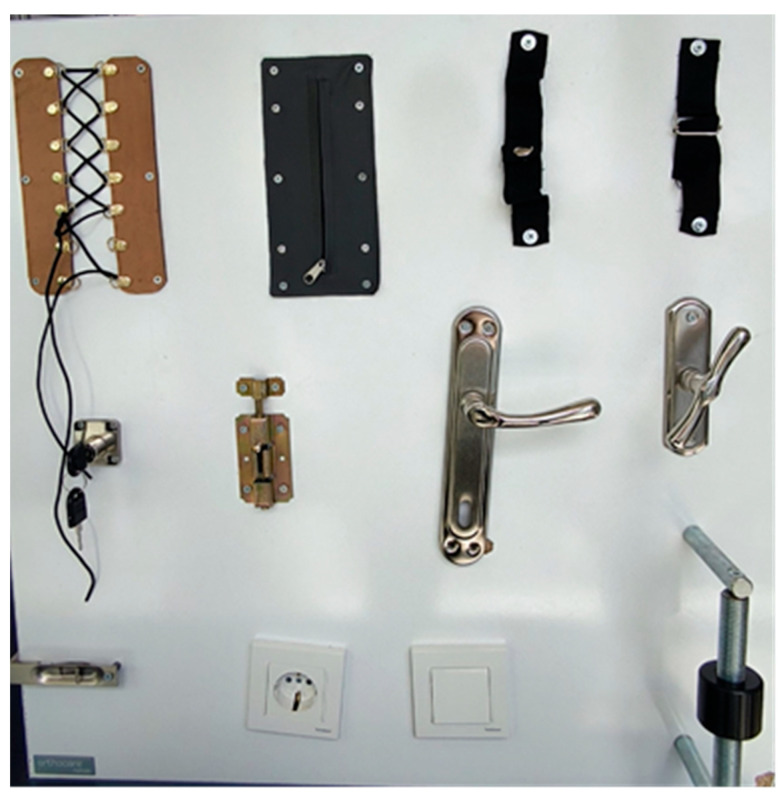
Functional panel simulating tasks of the Sollerman Hand Function Test.

**Table 1 diagnostics-15-01763-t001:** Distribution of participants by ethnic origin, sex, number of hands assessed, and mean age.

Ethnic Group	Sex	*n* (Individuals)	*n* (Hands)	Mean Age (Mean ± SD)
Turkish	Male	109	218	21.7 ± 2.0
Turkish	Female	103	206	21.3 ± 1.8
Georgian	Male	97	194	21.6 ± 2.1
Georgian	Female	91	182	21.2 ± 1.9
Total		400	800	21.4 ± 2.0

*n*: number of individuals, SD: standard deviation.

**Table 2 diagnostics-15-01763-t002:** Distribution of the palmaris longus muscle according to ethnic group.

Ethnic Group	PLM Present (*n*, %)	PLM Absent (*n*, %)	Total Hands (*n*)	χ^2^	*p* Value
Turkish	306 (72.2%)	118 (27.8%)	424	6.34	0.012
Georgian	301 (80.1%)	75 (19.9%)	376		
Total	607 (75.9%)	193 (24.1%)	800		

χ^2^: chi-square test, *p*: significance value.

**Table 3 diagnostics-15-01763-t003:** Distribution of the palmaris longus muscle by sex within ethnic origin groups.

Ethnic Group	Sex	PLM Present (*n*, %)	PLM Absent (*n*, %)	Total Hands	χ^2^	SD	*p*
Turkish	Male	161 (73.9%) ^a^	57 (26.1%) ^a^	218	7.88	3	0.048
	Female	145 (70.4%) ^a^	61 (29.6%) ^a^	206			
Georgian	Male	158 (81.4%) ^a^	36 (18.6%) ^b^	194			
	Female	143 (78.6%) ^a^	39 (21.4%) ^a^	182			
Total		607 (75.9%)	193 (24.1%)	800			

χ^2^: chi-square test, SD: standard deviation, *p*: significance value, ^a^,^b^: values with different superscript letters within the same column differ significantly (*p* < 0.05).

**Table 4 diagnostics-15-01763-t004:** Comparison of the effect of palmaris longus muscle absence type on upper extremity functions according to ethnic origin.

Ethnic Origin	PLM Absent Type	*n*	UEFI-15 (Mean ± SD)	U	*p*
Turkish	Unilateral	76	55.2 ± 2.3	719.5	0.442
	Bilateral	21	54.8 ± 2.6		
Georgian	Unilateral	49	55.6 ± 1.9	139.0	0.388
	Bilateral	13	55.1 ± 2.1		

U: Mann–Whitney U test, *n*: number of individuals, *p*: significance value.

**Table 5 diagnostics-15-01763-t005:** Effects of palmaris longus presence, ethnic origin, and sex on Sollerman test scores.

Variable	B (Coefficient)	Std. Error	z-Value	*p*-Value	95% CI (Lower–Upper)
Intercept	77.24	0.32	238.52	<0.001	76.61–77.88
PLM Presence	+0.33	0.29	1.15	0.249	−0.23–0.90
Ethnic Group (Georgian)	+0.26	0.25	1.03	0.302	−0.23–0.75
Sex	−0.08	0.25	−0.31	0.760	−0.57–0.41
Hand Side	−0.13	0.25	−0.51	0.608	−0.61–0.36

B: regression coefficient; Std. Error: standard error; z-value: test statistic; *p*-value: level of significance; 95% CI: 95% confidence interval.

## Data Availability

The data presented in this study are available on request from the corresponding author. The data are not publicly available due to privacy and ethical restrictions.

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
