# Peer review of "Investigation of the Effect of Palmaris Longus Presence on the Upper Extremity and Hand Functions in Individuals of Different Ethnic Origins"

_diagnostics, 2025, doi:10.3390/diagnostics15141763_

Round 1

Reviewer 1 Report

Comments and Suggestions for Authors

General comments:

Below is my review of the manuscript entitled “Investigation of the Effect of Palmaris Longus Presence on Upper Extremity and Hand Functions in Individuals of Different Ethnic Origins”.

This manuscript presents observational data comparing the prevalence of the palmaris longus (PL) muscle and its relationship to hand function across ethnic groups, with a specific focus on Georgian and Turkish individuals. The topic is clinically relevant, particularly for its implications in surgical graft planning. The study benefits from a clear objective and a substantial literature foundation, especially in discussing ethnic and anatomical variation.

However, the discussion could be improved in terms of clarity, organization, and critical analysis. At times, the narrative is overly descriptive and dense, with long passages of literature review that might be better synthesized or tabulated. Some statements about functional insignificance of PL lack sufficient statistical or methodological backing. The authors draw reasonable conclusions, but these should be framed more cautiously given the study’s limitations, including the cross-sectional design, reliance on clinical palpation without imaging confirmation, and a relatively homogeneous, healthy, and young sample. Additionally, the functional assessment tools used may not be sensitive enough to detect subtle motor contributions of PL.

Specific comments were listed below:

Line 31 – “can be observed in agenetic, paired, split, incomplete, digastric forms”. What is the clinical relevance of these forms? Explanation of their impact would be beneficial rather than just listing them.

Line 34 – Consider clarifying why this is frequently preferred.

Line 35 – “plays a supportive role in grip strength and hand functions”. This is a general statement that may need appropriate citations. This needs support since some studies show PL has minimal contribution to grip strength.

Line 47 – Be precise and cite original data for ethnic-based differences, especially in a U.S. context where methods and samples may vary.

Line 50-51 – What specific functions are limited (dexterity, strength, coordination)? This is an important gap, be more specific and more motivated.

Line 54 – Is there any hypothesis?

What is the activity level of these participants?

Line 61-70 – This classification method may introduce bias. Fluency in Georgian does not fully validate ethnic origin, especially in multiethnic or culturally mixed populations. Might need to acknowledge the potential for misclassification bias and explain how it was mitigated.

Line 96 – UEFI-15 evaluates bilateral function, how was this linked to unilateral PL absence? If the UEFI-15 doesn't differentiate sides, clarify how asymmetrical PL presence was interpreted in the analysis.

Line 143 – Clarify whether the 800 hands are independent in analyses or if dependency is considered.

Line 149 - “Etnik Group” should be corrected to "Ethnic Group."

Line 151-155 – This is just a prevalence comparison; the biological or cultural explanation for the difference is absent.

What are superscripts a and b referring to in Table 3? Need to legend to clarify this.

Line 186 – The section opens clearly with a summary of the key findings. However, the phrasing "the most important finding" (line 185) could be made more specific and objective (e.g., “The primary outcome observed was…”). Additionally, the distinction between statistical significance and practical relevance could be elaborated. For example, while a significant ethnic difference in PL prevalence is reported, it is not discussed whether this has clinical or surgical relevance.

Line 195 – The conclusion that PL is a “safe graft candidate” because of its lack of functional contribution is reasonable but would benefit from citing specific studies or surgical outcomes where PL grafting was successful and complication-free.

Line 279-297 – The functional implications section is appropriate, but the authors should be cautious in overstating the lack of function. Some studies show minimal but present effects on fine motor tasks. Consider rephrasing to reflect that PL absence may not significantly impair function, especially in a healthy population.

The limitation could be expanded specifically:

  • The lack of imaging validation is a major methodological limitation. Consider recommending ultrasound or MRI in future work.
  • Occupational hand use is an important variable, especially in populations with diverse labor profiles. This deserves stronger emphasis.
  • The cross-sectional design precludes conclusions on long-term hand function; this should be explicitly stated as limiting causal inference.

Author Response

Dear reviewer, thank you for carefully reviewing our work and sharing your opinions. We believe that your comments will further strengthen our work. In this context, the requested changes have been made and are explained below item by item.

Line 31—The clinical significance and effects of the forms provided were explained.

Line 34—Frequency of use was added.

Line 35—A source was added to strengthen the study.

Line 47—A US-based study was added.

Lines 50–51—Additions related to functions were made.

Line 54—The study hypothesis was added.

Lines 61-70: Details regarding classification were added, and measures taken to reduce bias were included.

Line 143: Dependent-independent evaluation conditions related to the 800 hands used in the study were added to the study.

Line 149: Spelling errors were corrected.

Satır 186: Çalışmanın tartışma bölümü yeniden yapılandırıldı, istenen yazım değişiklikleri yapıldı, yaygınlık ve cerrahi önemi anlatıldı.

*Tablo 3'te verilen semboller için açıklamalar eklenmiş ancak açıklanmamıştır.

*Çalışmada görüntüleme yöntemlerinden yararlanılması gerektiği anlatıldı.

Reviewer 2 Report

Comments and Suggestions for Authors

Thank you for the opportunity to review the article entitled "Investigation of the Effect of Palmaris Longus Presence on Upper Extremity and Hand Functions in Individuals of Different Ethnic Origins". This article was described a well-executed study of several different ethnic populations (Georgian and Turkish) as well as sex differentiations regarding the presence and appearance of the palmaris longus muscle. Please note that the use of gender in this article is incorrect. It should be "sex" since you are not describing how each subject identifies (gender) but rather than assigned sex at birth. I also suggest the use of PLM rather than PL and FCRM rather than FCR. Other than these and a few other minor grammatical suggestions, I think this is an incredibly interesting study and manuscript. I'm excited to hear about its future studies into other populations and their comparisons to one another.

Author Response

Dear reviewer, first of all, we would like to thank you for evaluating our work. Your comments on our work have also motivated us, and we are very grateful. As requested, corrections have been made to the word “gender.” In addition, abbreviations have been checked and corrected. Grammar has also been checked again. Thank you very much once again.

Reviewer 3 Report

Comments and Suggestions for Authors

The manuscript aims to investigate the presence and absence of palmaris  longus.

A topic that it is already extensively studied in the current literature. 

The reviewer is not sure that the assessment according to functional tests can be 100% accurate. 

The ultrasonography (US) is the most appropriate method according to most of the studies.

Nevertheless, the authors have not presented results with presence and absence of the muscle with adequate figures.

Important references are missing, such as the current meta-analysis on the topic.

Author Response

Dear reviewer, first of all, thank you for evaluating our work. The requested corrections have been made and the updated version of the article has been uploaded.

Round 2

Reviewer 1 Report

Comments and Suggestions for Authors

The authors have addressed all my concerns. Thank you. 

Author Response

Sayın yorumcu, çalışmamıza katkılarınız için teşekkür ederiz.

Reviewer 2 Report

Comments and Suggestions for Authors

Unfortunately, there are many areas in the revised texts where PLM was not inserted. PL muscle needs to be changed to PLM, PL needs to be changed to PLM, or palmaris longus muscle needs to be changed to PLM on all of the following lines: 58, 73, 80, 138, 144-145, 148, 149-150, 496-497, 500, 505, 515-516, 517, 519, 522, 528, 532, 675 (PLM tendon), 676, 683, 684-685, 689, 691, 693, 698, 705, 822, 824-825, 831, 834 (PLM tendon), 842, 851, 852, 853, 857, 861, 946, 954, 958, 960, 961, and 964. I would also recommend changing musculus flexor carpi radialis to flexor carpi radialis muscle and m. flexor carpi ulnaris to flexor carpi ulnaris muscle (lines 78 and 79). Caucasians should be changed to Whites in line 140. Muscle needs to be added after flexor digitorum superficialis on line 678. FCR had not been defined before line 708 so it needs to be put in on line 78. I would change line 844-845 to flexor digitorum superficialis muscle (FDSM) to align with the PLM edits.

Author Response

Dear reviewer, thank you very much for your review. All requested changes have been made in accordance with your comments.

In addition, changes have been made to musculus flexor carpi radialis to flexor carpi radialis muscle and m. flexor carpi ulnaris to flexor carpi ulnaris.

Kafkasyalılar Beyazlar olarak değiştirildi.

FCR kasında değişiklikler yapıldı.

FDS kası FDSM olarak değiştirildi.

Reviewer 3 Report

Comments and Suggestions for Authors

Thank you for revising according to the reviewers' comments.

Author Response

Dear reviewer, thank you for your contribution to our study.